# Prevalence and Factors Associated with Waterborne Diseases in Couffo, Southwestern Benin: The Case of Aplahoué

**DOI:** 10.3390/ijerph22010058

**Published:** 2025-01-03

**Authors:** Cyriaque Degbey, Eunice Houessionon, Christophe de Brouwer

**Affiliations:** 1Department of Environmental Health, Regional Institute of Public Health, University of Abomey Calavi, Cotonou 01 BP 918, Benin; 2University Clinic of Hospital Hygiene, National University Hospital Center Hubert Koutoukou Maga, Cotonou 01 BP 386, Benin; 3National Water Institute, University of Abomey-Calavi, Cotonou 01 BP 526, Benin; eunicehouessionon@gmail.com; 4Occupational Health Unit and Environmental Toxicology, School of Public Health, Free University of Brussels, 1070 Brussels, Belgium; de.brouwer.christophe@ulb.ac.be

**Keywords:** waterborne diseases, prevalence, health education, Benin

## Abstract

Water-related diseases are among the infectious diseases that represent a major public health challenge in developing countries. This study aimed to assess the prevalence of waterborne diseases and the factors associated with their occurrence in the commune of Aplahoué, located in southwestern Benin. A cross-sectional analytical study was conducted among 125 households selected through simple random sampling in the commune. Data analysis was performed using regression models and Chi-square tests with SPSS version 27.0, considering a *p*-value below 5% as statistically significant. This study revealed an overall prevalence of waterborne diseases of 45.6% at the household level and 16.6% at the individual level. Malaria was the most prevalent waterborne disease (8.4%), followed by diarrhea (6.3%). Multivariate analysis identified a history of waterborne diseases and hospitalization due to waterborne diseases as significant independent risk factors for the occurrence of new waterborne diseases. It is, therefore, critical to improve access to safe drinking water, promote better hygiene and sanitation practices, and strengthen health education through effective communication to foster behavioral change within the community.

## 1. Introduction

In developing countries, particularly in sub-Saharan Africa, nations face challenges regarding access to safe drinking water and sanitation. Despite significant progress over the past decades, the World Health Organization (WHO) estimated that in 2019, 319 million people in sub-Saharan Africa still lacked access to improved drinking water sources. Furthermore, access to adequate sanitation facilities remains limited, leading to water contamination and the spread of waterborne diseases [1,2]. Consequently, communities are highly exposed to the risk of contracting waterborne illnesses. Diarrhea, cholera, and malaria represent the most pressing water-related health concerns in the region, disproportionately affecting the most vulnerable populations, particularly children, the elderly, and residents of rural areas or informal settlements [3]. Clearly, the prevalence of waterborne diseases in sub-Saharan Africa raises significant concern among residents, policymakers, healthcare professionals, and researchers.

In line with other sub-Saharan African countries, Benin also grapples with issues surrounding water, sanitation, and hygiene (WASH) across both rural and urban settings. While remarkable progress has been made in urban and rural drinking water access, reaching 77% and 66%, respectively, improvements in sanitation services have been marginal [4]. Disparities persist across regions, with over half of the population lacking access to improved sanitation facilities [2]. This situation exposes communities to various forms of waterborne and water-related diseases. Indeed, Benin continues to experience a high prevalence of water-related diseases that contribute significantly to mortality and morbidity. Health statistics from 2021 revealed that waterborne diseases accounted for 52.3% of all illnesses in Benin [5]. The most prevalent include malaria, diarrhea, typhoid fever, and schistosomiasis.

Several researchers have explored the prevalence and causes of waterborne diseases in southern Benin, specifically within the communes of Abomey-Calavi, Toffo, Zè, Lokossa, and Lalo [6,7,8,9,10]. In Aplahoué, the largest and most populated commune in Couffo (a region in southwestern Benin), only 37.4% of the population has access to drinking water, while improved sanitation coverage remains as low as 14.6% [11]. This situation aligns with findings from a 2017 project conducted as part of a decentralized cooperation initiative between Ramonville Saint-Agne and Aplahoué aimed at constructing and rehabilitating water and sanitation infrastructure. The study highlighted that most residents of Aplahoué, except those in the main town and certain primary schools, lack access to sanitation facilities. Additionally, water access conditions have deteriorated over the years, with faulty and insufficient Village Water Supply (AEV) systems making safe water access increasingly challenging. As a result, many residents rely on water from ponds, wells, and household tanks, limited access to which often results in waterborne illnesses. To date, no studies have focused on waterborne diseases in this commune despite limited water and sanitation access, which suggests that such diseases may indeed be prevalent.

This study thus focuses on Aplahoué commune to assess its water-related health situation. 

## 2. Materials and Methods

### 2.1. Study Setting

This study was conducted in Aplahoué, in southern Benin, covering an area of 915 km^2^. Agriculture is the dominant economic activity, engaging 56% of the active population. According to the latest census, in 2013, Aplahoué had a population of 171,109, up from 116,988 in 2002, reflecting an intercensal growth rate of 3.42% [12]. Access to safe drinking water remains a significant challenge for residents, with only 37.5% of households having reliable access, according to RGPH-4 [12]. Potable water scarcity also persists, especially in urban areas. The Benin Water Company (SONEB) primarily serves the urban population of Aplahoué through private subscriptions. However, in neighborhoods not connected to SONEB, some households rely on private wells, boreholes, or informal vendors who typically serve the poorest households. Figure 1 presents the geographical location of the commune of Aplahoue.

### 2.2. Study Design

This was a cross-sectional, descriptive, and analytical study. Data were collected from the population between 4 September and 16 October 2023. 

### 2.3. Study Population

This study targeted households in the commune of Aplahoué. Household heads who provided informed consent and had resided in the area for at least six months were included. Those who declined to cooperate or to provide informed consent were excluded.

### 2.4. Sampling

The commune of Aplahoué was divided into districts, which were treated as clusters. We obtained a list of villages within each district of the commune. For each district, we visited the villages and their hamlets. Upon arriving in each village, we divided it into four quadrants using the cardinal points technique to ensure complete coverage. In each quadrant, households were selected by skipping every five households to ensure a balanced distribution of surveyed households within the village. In the first house encountered, the head of the household or their representative was interviewed. If no one was found in the first household, we proceeded to interview the head or representative of the next household. This process was repeated until all households in the district were surveyed and all quadrants were covered. To maintain fairness in the distribution of households within each district, a weighting method was applied based on the total number of households in each district. The sample size of household heads surveyed was determined using Schwartz’s formula [13], considering a risk α = 5%, a prevalence *p* = 8%, and a precision *i* of 5%. The estimated sample size was 114 households. An additional 10% margin was added to account for non-respondents, bringing the total number of household heads enrolled to 125.

### 2.5. Variables

#### 2.5.1. Dependent Variables

The dependent variable was the occurrence of waterborne diseases. This dichotomous variable was coded as 1 if any household member had experienced a waterborne disease at least once during the three months preceding the survey and 0 otherwise. Data on this variable were collected based on declarations from the head of the household or their representative. The waterborne diseases considered in this study included malaria, diarrhea, dysentery, typhoid fever, and skin infections (such as scabies).

#### 2.5.2. Independent Variables 

The independent variables in this study were grouped into four broad categories: Demographic factors: sex (male, female), area of residence (rural, urban), age in years (18–30, 31–59, 60 and over), household size (1–3, 4–6, 7 and over), ethnicity (adja, fon, mina), level of education (no schooling, primary, secondary, higher), and occupation (farmer, craftsman, other, shopkeeper, civil servant, housewife, private worker);Behavioral factors: main source of water (rainwater, run-off water, boreholes, wells, rivers, soneb), treated and safe source (no, yes), and water covered during transport (no, yes);Environmental factors: availability of toilets (no, yes) and presence of stagnant water (no, yes);Health factors: knowledge of waterborne diseases (no, yes), history of waterborne disease (no, yes), treatment at health center (no, yes), frequency of occurrence of illness (none, 1 time, 2 times, several times), use of mosquito net (no, yes), hospitalization due to waterborne disease (no, yes), and training or information received on prevention (no, yes).

### 2.6. Data Collection Techniques and Tools

Two data collection techniques were employed: survey questionnaires and direct observation. The survey involved the use of a structured questionnaire featuring open-ended questions aimed at identifying household members in the commune of Aplahoué and providing basic information about them. This questionnaire was developed using Google Forms, an online tool that facilitates the creation of surveys, questionnaires, and forms. The questionnaire was primarily addressed to the household head (male or female) or, in their absence, to another household member aged 18 years or older. Direct observations were conducted during the household surveys to gather data on physical and behavioral risk factors at the household level, assess sanitation and general hygiene, describe the environment surrounding water supply infrastructure, and examine the means and methods used by households to transport drinking water. Observations were recorded using observation sheets and notebooks to document risk factors and behaviors. Photographs were also taken to provide visual documentation of household sanitary conditions. The data collection tools included data extraction sheets, the household questionnaire, an interview guide for authorities, and an observation sheet. A pretest was conducted to evaluate and refine the questionnaire. The pretest took place in the commune of Djakotomey. Data collection was carried out over a one-month period during September 2023.

In addition to collecting data by questionnaire, we consulted the health statistics yearbooks of the Aplahoue health zone from 2019 to 2022 to identify water-related diseases prevalent in the municipality. A cross-referencing was performed with the questionnaire data to ensure the veracity of household declarations. A data collection sheet was used to extract and organize the data.

### 2.7. Data Analysis

After collecting survey responses via Google Forms, we conducted a data processing phase involving the extraction, organization, and verification of data in preparation for statistical analysis. The collected data were analyzed using SPSS (Statistical Package for the Social Sciences) version 27.0. The frequency distribution of each variable was calculated to understand their patterns within the dataset. The prevalence of waterborne diseases was computed, and the resulting frequencies and tables were entered into Excel to produce graphical representations. Pearson’s Chi-square test with a significance threshold of 5% and the relative risk (Odds Ratio) was employed to identify associations between factors and waterborne diseases. Variables with a p less than 0.02 in the univariate analysis were included in the logistic regression model using the stepwise forward Wald method. Final results are presented in a summary table, including the Odds Ratios (OR), 95% confidence intervals (95% CI), and *p*.

### 2.8. Ethical Approval

This study received approval from the Departmental Health Directorate of Couffo. Informed consent was obtained from all participants before the interviews. Participants were free to withdraw at any time without any repercussions. Anonymity was strictly maintained, and the results were kept confidential. Participation in the survey was entirely voluntary, and no household was coerced into participating, ensuring respect for participants’ autonomy. During each household visit, the data collection authorization letter was presented, reinforcing the transparency of this study and demonstrating that it had received official approval and was conducted ethically.

## 3. Results

### 3.1. Basic Characteristics of Households 

The sample of this study consisted of 125 households. It was composed of 60% male household heads, with a sex ratio (M/F) of 1.5 (Table 1). A total of 34.4% of household heads were non-literate compared to those who were educated. Among the educated, 37.6% had a primary level of education, 20% a secondary level, and a minority (8%) held higher education degrees. The most frequently reported income-generating activities were agriculture (26.4%). Among the 125 surveyed households, 22.4% were small households, 32% were medium-sized households, and 45.6% were large households.

### 3.2. Prevalence of Waterborne Diseases

The prevalence of waterborne diseases was 45.6%. The most frequent waterborne diseases were malaria (8.4%) and diarrhea (6.3%). Figure 2 shows the prevalence of these diseases.

### 3.3. Univariate Analysis

#### 3.3.1. Socio-Demographic Characteristics

Regarding sex (Table 1), women were significantly more likely to suffer from waterborne diseases than men (64.0% vs. 33.3%, *p* = 0.0007). This prevalence decreased with increasing levels of education: primary (44.7%, *p* = 0.05), secondary (32.0%, *p* = 0.008), and higher education (0%). Households with from 4 to 6 members had a prevalence of 47.6% (*p* = 0.009), and those with 7 or more members had a prevalence of 60.8% (*p* = 0.0001), compared to households with from 1 to 3 members (18.8%). Regarding occupation, farmers had the highest prevalence of waterborne diseases (63.6%). Other professions showed varying prevalence rates: craftsmen (54.6%, *p* = 0.50), traders (40.0%, *p* = 0.07), and private employees (20.0%, *p* = 0.005).

#### 3.3.2. Behavioral Characteristics

Households relying on surface water and rivers as their primary water source had a 100% prevalence of waterborne diseases (*p* = 0.007 and *p* = 0.001, respectively) (Table 1). Households using wells also had a high prevalence of 54.2% (*p* = 0.01). In contrast, households using water from SONEB (the national water utility) had a much lower prevalence of 10%. Households using treated and safe water sources had a waterborne disease prevalence of 34.2%, compared to 63.3% for those using untreated sources (*p* = 0.001).

#### 3.3.3. Environmental Characteristics

Households without toilets had a prevalence of waterborne diseases of 69.4%, compared to 30.3% for those with toilets (*p* < 0.001) (Table 1). Households with stagnant water within their compounds had a prevalence of 77.1%, compared to 33.3% for those without stagnant water (*p* < 0.001). In terms of hygiene practices, households where members washed their hands before handling drinking water had a prevalence of 9.5%, compared to 63.9% for those who did not (*p* < 0.001).

#### 3.3.4. Sanitation Practices

Households with a history of waterborne diseases had a prevalence of 64.7%, compared to 5.0% for those without such a history (*p* < 0.001) (Table 1). Households where members had received treatment at a health center had a prevalence of 82.4%, compared to 31.9% for those who had not (*p* < 0.001). Households with multiple occurrences of waterborne diseases had a prevalence of 64.2% (*p* < 0.001), and those with two occurrences had a prevalence of 60.0% (*p* = 0.0003), compared to 5.3% for households without any occurrence. Households with members hospitalized due to waterborne diseases had a prevalence of 84.8%, compared to 31.5% for those without hospitalizations (*p* < 0.001).

### 3.4. Multivariate Analysis

After adjustment, logistic regression modeling identified factors associated with the occurrence of waterborne diseases among households in the commune of Aplahoué (Table 2). The analysis revealed that households with a history of waterborne diseases had a significantly higher risk of experiencing new episodes of waterborne diseases (OR = 16.99, *p* = 0.006). Similarly, households with members hospitalized due to waterborne diseases also had a significantly higher risk of experiencing new episodes (OR = 16.39, *p* = 0.04). These findings indicate that a history of waterborne diseases and hospitalization due to waterborne diseases are significant independent risk factors for the occurrence of new episodes. This suggests that households previously affected by waterborne diseases are more likely to experience further episodes.

## 4. Discussion

In this section, we delve into an in-depth analysis of the findings from our study. We assess the achievement of our goals, evaluate the quality and validity of our results, and compare them to existing data in the scientific literature.

The objective of this study was to examine the prevalence and factors associated with the occurrence of waterborne diseases in the commune of Aplahoué. By combining various data collection techniques, we were able to describe common waterborne diseases in Aplahoué, calculate their prevalence, and identify factors associated with their occurrence. Based on our results, we consider that the study objectives were successfully achieved.

Our study adopted a cross-sectional, descriptive, and analytical approach. The quality of the data depended on our own expertise as investigators. An information bias may have been introduced because the data were obtained by declaration from the respondents, which may have posed problems of overestimation, underestimation, or even trivialization of these questions by certain respondents. Confounding bias could be minimized using binary logistic regression. Although the sample size is relatively modest, which could reduce the statistical power of this study and make it more difficult to detect real effects, it nevertheless made it possible to carry out an in-depth analysis of the data collected. Despite these limits or inadequacies, these do not affect the credibility and originality of this study. The quality of the results is therefore valid.

The prevalence of waterborne diseases is high in the study area, but we were unable to study the microbiological quality of drinking water in households in the study area, which did not allow us to assert that exposure to contaminated water is the main cause of these diseases, which constitutes a limitation for our study. No study has been conducted in the study area to prove that the presence of these diseases is due to the consumption of contaminated water.

Our findings revealed an overall prevalence of waterborne diseases of 45.6% at the household level. Nearly half of the surveyed households in Aplahoué are affected by waterborne diseases, indicating that many households face significant risks of water-related illnesses. A more detailed analysis shows that malaria is the most prevalent condition. This result aligns with a prevalence of 8.7% observed in Ouagadougou in 2014, though it is 1.15 times lower than the 10% prevalence recorded in Cameroon [14,15,16]. These variations could be attributed to the timing of the surveys, which were conducted during the rainy season, as well as the widespread use of mosquito control measures by the population. Additionally, differences in climate, environmental conditions, and geography between study areas might explain these discrepancies.

The study shows that the age group most vulnerable to malaria conditions is those under 5 years old. This observation is the same as that made by Somé et al. in 2014 [14] and M. Rostant in 2022 [15].

Diarrhea, another significant condition, exhibited a general prevalence of 6.3%. This figure is significantly lower than the 20% and 29% prevalences reported by Safougne et al. in 2020 [16], respectively. Several factors could explain these differences. First, the populations studied differ: our sample consists of specific households in Aplahoué, while the other studies focused on children in Ethiopia and residents of urbanized mangrove areas in Cameroon. Furthermore, the environmental contexts vary substantially between these regions, particularly regarding access to potable water, sanitation, and hygiene practices.

From the point of view of drinking water supply, it should be noted that households in the commune of Aplahoué use various sources of water, which can have direct consequences on the quality of the water they consume. Indeed, runoff and rivers are sources often exposed to contamination by fecal matter, chemicals, and waste. The same observation was made by Mialo et al. in 2016 in the commune of Lalo in South Benin [10]. Our study highlighted crucial aspects concerning the transport and management of water by households in the commune of Aplahoué. The majority of households (78.4%) carry water from the source. This shows that few households have a water point at home. Cans and basins are the most commonly used containers. Long journeys can cause fatigue and increase the risk of water contamination during transport. These same observations were made by Somé et al. in 2014 [14] and Mialo et al. in 2016 [10]. When the time required to collect water exceeds a few minutes (approximately five minutes or at a distance of 100 meters from the residence). Regarding water storage, 25-liter drums, and jars are the most commonly used containers. They can limit the amount of water stored if they do not have large capacities. It is important to emphasize that repeated opening and closing of containers during use exposes them to the environment. This can lead to a change in water quality, exposing users to health risks. This observation is similar to that made by Todedji J in 2013 in Benin, who, in his study, stated that the storage of water in the household leads to a deterioration of water quality due to its recontamination in the house [17]. Several results from previous studies confirm that water can gradually become contaminated when stored and transported. In 2018, Ahouansè et al. [18] noted progressive contamination of the water from the standpipe between transport and storage. The analysis of hygiene-related behaviors within households in the commune of Aplahoué reveals that children and the majority of adults do not wash their hands before handling drinking water.

The multivariate analysis highlighted that a history of waterborne diseases and hospitalization due to such illnesses are significant independent risk factors for the occurrence of new waterborne diseases. This complex analysis confirmed the importance of these two factors by showing that they independently contribute to an increased risk of new infections, even after adjusting for other variables. This finding underscores that individuals with a history of waterborne diseases or hospitalizations related to these conditions form high-risk groups. Our results align with those of previous studies, including Hounkpè et al. (2018) and Smith et al. (2017), which also identified a history of waterborne diseases as a critical risk factor, even after controlling for confounding variables [19,20]. While our findings are broadly consistent with those of Hounkpè et al. (2018) and Smith et al. (2017), it is noteworthy that our study also accounted for variables specific to the Aplahoué region, such as the availability of sanitation infrastructure, personal hygiene practices, income levels, education, access to healthcare, and the management of solid and liquid waste [19,20]. These region-specific variables might explain some of the minor differences observed in the risk coefficients. 

A study conducted in Kenya by Mwangi et al. in 2018 found that households using unsecured water sources, like rivers and wells, had a higher prevalence of waterborne diseases, which is consistent with our results [21]. It was noted that households using a safe, treated water source had a lower prevalence of waterborne diseases than those using an untreated source. Water treatment at home, such as filtration or chlorination, can, therefore, significantly reduce the risks of waterborne diseases [22]. A study in India showed that households using home water treatment methods had a lower prevalence of waterborne diseases, supporting our conclusions on the importance of treated water [23]. Other studies have found similar results, suggesting that water coverage during transport may not be sufficient to prevent contamination [24]. The one conducted by Gebre et al. in 2017 in Ethiopia also confirmed that water coverage during transport did not have a significant impact on the prevalence of waterborne diseases, which is in agreement with our results [25].

Our study employed a cross-sectional, descriptive, and analytical approach. Information bias may have arisen due to participants’ tendency to respond "correctly" rather than truthfully about their daily realities. We mitigated this bias by encouraging participants to provide honest responses. Although the sample size was relatively modest, it allowed us to conduct a thorough analysis of the collected data.

## 5. Conclusions

Our findings underscore the importance of targeting identified high-risk groups through specific waterborne disease prevention strategies. These strategies could include tailored health education programs, regular medical follow-up, improved access to treated water sources, and adequate sanitation facilities. Strengthening health infrastructure to ensure the timely and appropriate treatment of waterborne diseases could also contribute to reducing their prevalence. Future research should explore other region-specific factors that might influence these associations and evaluate the effectiveness of targeted interventions for these groups.

## Figures and Tables

**Figure 1 ijerph-22-00058-f001:**
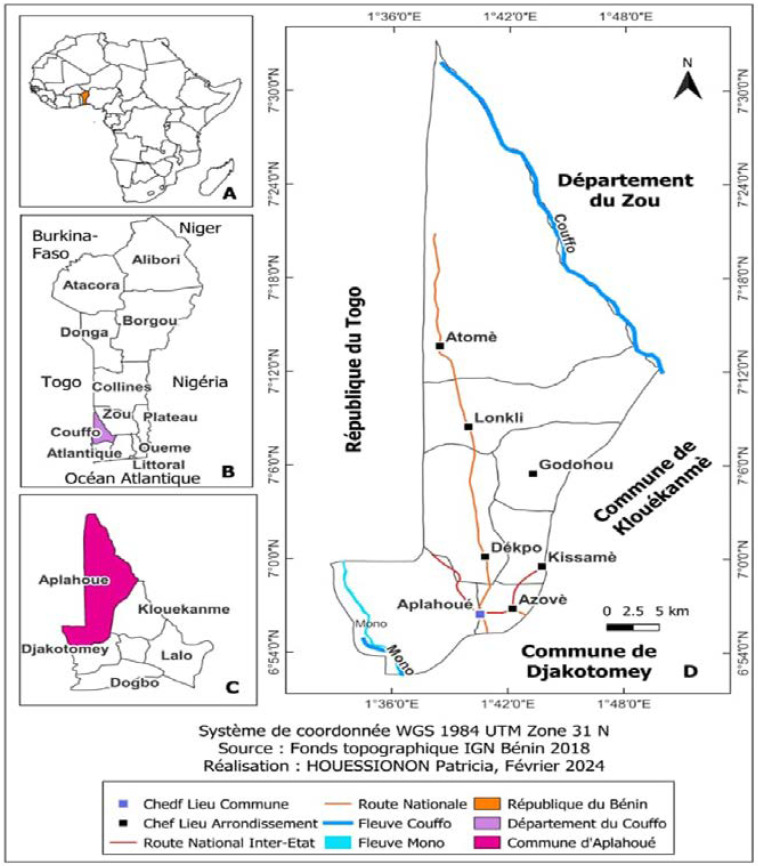
Geographical location of the commune of Aplahoué.

**Figure 2 ijerph-22-00058-f002:**
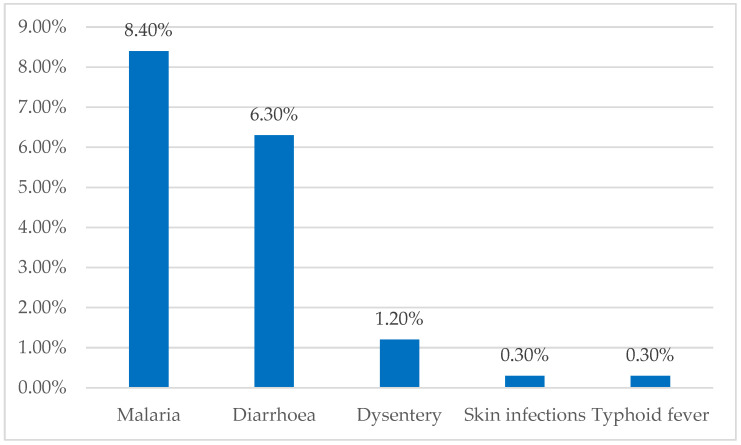
Prevalence of water-related diseases in Aplahoué.

**Table 1 ijerph-22-00058-t001:** Univariate analysis of health factors in waterborne diseases.

Variables	*n*	%	Waterborne Disease = Yes				
			*n*	%	OR	95% CI	*p*
**Sociodemographic characteristics**							
**Sex**							
Male	75	60.0	25	33.3	1	-	
Female	50	40.0	32	64.0	3.55	1.6–7.5	0.0007 **
**Area of residence**							
Rural	34	27.2	45	49.5	1		
Urban	91	72.8	12	35.3	0.55	0.2–1.3	0.15
**Age (years)**							
18–30	11	8.8	9	64.3	1	-	
31–59	108	86.4	46	43.8	0.43	0.13–1.4	0.14
60 and over	6	4.8	2	33.3	0.27	0.03–2.1	0.20
**Household size**							
1–3	28	22.4	6	18.8	1	-	
4–6	40	32.0	20	47.6	3.9	1.3–11.5	0.009 *
7 and over	57	45.6	31	60.8	6.7	2.3–19.2	0.0001 **
**Ethnicity**							
Adja	111	88.8	52	47.3	1	-	
Fon	10	8.0	4	40.0	0.7	0.2–2.7	0.65
Mina	5	3.2	1	20.0	0.3	0.03–2.5	0.23
**Level of education**							
No schooling	47	37.6	28	65.1	1	-	
Primary	25	20.0	21	44.7	0.43	0.2–1.01	0.05
Secondary	10	8.0	8	32.0	0.25	0.08–0.7	0.008 *
Higher	43	34.4	0	0.0	-	-	-
**Occupation**							
Farmer	33	26.4	21	63.6	1	**-**	
Craftsman	8	6.4	12	54.6	0.68	0.22–2.05	0.50
Other	22	17.6	2	14.3	0.09	0.02–0.5	0.001 *
Shopkeeper	5	4.0	10	40.0	0.38	0.13–1.10	0.07
Farmer	3	2.4	6	75.0	1.71	0.29–9.86	0.54
Civil servant	25	20.0	1	20.0	0.14	0.01–1.4	0.06
Housewife	15	12.0	2	66.7	1.14	0.09–13.9	0.91
Private worker	14	11.2	3	20.0	0.14	0.03–0.6	0.005 *
Behavioral characteristics							
**Main source of water**							
Rainwater	13	10.4	6	46.2	7.7	0.7–79.8	0.06
Run-off water	2	1.6	2	100.0	-	-	0.007 *
Boreholes	45	36.0	15	33.3	4.5	0.5–38.9	0.14
Wells	48	38.4	26	54.2	10.6	1.2–90.6	0.01 *
Rivers	7	5.6	7	100.0	-	-	0.001 *
SONEB	10	8.0	1	10.0	1	-	
**Treated and safe source**							
No	49	39.2	31	63.3	1	-	
Yes	76	60.8	26	34.2	0.30	0.14–0.63	0.001 **
**Water covered during transport**							
No	94	75.2	42	46.2	1	-	
Yes	31	24.8	15	48.4	1.16	0.5–2.6	0.71
Environmental characteristics							
**Availability of toilets**							
No	49	39.2	34	69.4	1	-	
Yes	76	60.8	23	30.3	0.19	0.08–0.4	<0.001 **
**Presence of stagnant water**							
No	90	72.0	30	33.3	1	-	
Yes	35	28.0	27	77.1	6.75	2.7–16.6	<0.001 **
**Health characteristics**							
**Knowledge of waterborne diseases**							
No	37	29.6	20				
Yes	88	70.4	37		0.61	0.3–1.3	0.21
**History of waterborne disease**							
No	40	32.0	2	5.0	1		
Yes	85	68.0	55	64.7	34.8	7.9–154.5	<0.001 **
**Treatment at health centre**							
No	91	72.8	29		1		
Yes	34	27.2	28		9.97	3.7–26.7	<0.001 *
**Frequency of occurrence of illness**							
None	38	30.4	2	5.3	1	-	
1 time	1	0.8	0	0.0	0.0	-	0.81
2 times	5	4.0	3	60.0	27.0	2.7–265.7	0.0003 **
Several times	81	64.8	52	64.2	32.3	7.2–143.9	<0.001 *
**Use of mosquito net**							
No	29	23.2	10		1	-	
Yes	96	76.8	47		1.88	0.8–4.3	0.17
**Hospitalization due to waterborne disease**							
No	92	73.6	29		1		
Yes	33	26.4	28		12.2	4.3–34.7	<0.001 **
**Training or information received on prevention**							
No	79	63.2	33		1		
Yes	46	36.8	24		1.5	0.7–3.2	0.26

* *p* ˂ 0.05; ** *p* ˂ 0.001.

**Table 2 ijerph-22-00058-t002:** Multivariate analysis of health factors in waterborne diseases.

Variables	OR	95% CI	*p*
**Sex**			
Male	1	-	
Female	1.11	0.17–7.09	0.91
**Level of education**			
No schooling	1	-	
Primary	0.70	0.11–4.16	0.69
Secondary	0.82	0.07–8.69	0.87
Higher	0	-	0.99
**Household size**			
1–3	1		
4–6	2.15	0.26–17.41	0.47
7 and over	1.31	0.18–9.28	0.78
**Occupation**			
Farmer	1		
Craftsman	0.59	0.08–4.24	0.60
Other	0.08	0.003–2.37	0.14
Shopkeeper	1.24	0.07–20.34	0.87
Farmer	3.29	0.05–200.29	0.56
Civil servant	0.54	0.00–75,352.9	0.91
Housewife	0.27	0.004–17.86	0.54
Private worker	0.07	0.004–1.66	0.10
**Main source of water**			
Rainwater	0.60	0.11–32.28	0.80
Run-off water	109,107,654.10	-	0.99
Boreholes	0.38	0.15–9.88	0.56
Wells	2.71	0.12–59.81	0.52
Rivers	324,870,315.58	-	0.99
SONEB	1	-	
**Treated and safe source**			
No	1		
Yes	1.64	0.18–14.65	0.65
**Availability of toilets**			
No	1		
Yes	0.93	0.13–6.45	0.94
**Presence of stagnant water**			
No	1		
Yes	2.27	0.47–15.58	0.26
**Wash hands before handling drinking water**			
No	1		
Yes	0.17	0.02–1.49	0.11
**History of waterborne disease**			
No	1		
Yes	16.99	1.88–324.91	0.006 *
**Treatment at health centre**			
No	1		
Yes	1.10	0.10–11.77	0.93
**Frequency of occurrence of illness**			
None	1		
1 time	2.36	-	1.00
2 times	4.96	0.03–672.45	0.52
Several times	1.16	0.05–25.85	0.92
**Hospitalization due to waterborne disease**			
No	1		
Yes	16.39	1.06–251.78	0.04 *

Note: * *p* ˂ 0.05; ** *p* ˂ 0.001

## Data Availability

The data used in this study can be obtained by sending a request to the authors.

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
