# Peer review of "Prevalence and Factors Associated with Waterborne Diseases in Couffo, Southwestern Benin: The Case of Aplahoué"

_ijerph, 2025, doi:10.3390/ijerph22010058_

Round 1

Reviewer 1 Report

Comments and Suggestions for Authors

Very good and interesting research. Please find my additional comments in the attachment.

Author Response

  1. Main Question Addressed by the Research

The main question guiding the research is:

What is the prevalence of waterborne diseases, and what are the factors associated with their occurrence in Aplahoué, Couffo region, Benin?

The objective is to determine the magnitude of the problem of waterborne diseases and identify the demographic, behavioral, environmental, and health-related variables influencing their occurrence.

The topic investigated is not original, as waterborne diseases have been widely studied, especially in developing countries. However, the study is highly relevant, particularly in rural contexts like Benin, where these diseases remain a significant cause of morbidity and mortality in Sub-Saharan Africa.

The study's originality lies in its focus on Aplahoué, a locality with limited water and sanitation infrastructure that had not been previously evaluated in depth. This local perspective fills a specific gap in information about health conditions and water access in the region.

Furthermore, the study confirms findings from broader literature, identifying risk factors and linking behaviors and environmental conditions to disease prevalence. These results can serve as a foundation for public policy interventions and comparative studies with other contexts in Sub-Saharan Africa.

These aspects justify its publication under the category of case studies.

Thank you for your comment.

  1. Does It Address a Specific Gap in the Field?

The study does not represent a significant advance in the state of the art on waterborne diseases, as it confirms previously established findings on the determinants and prevalence of these diseases in developing countries. The problem has been extensively studied in Sub-Saharan Africa and other regions with similar conditions (e.g., Cameroon, Burkina Faso, Ethiopia, and parts of Asia and Latin America). These studies have documented risk factors such as lack of access to safe drinking water, poor sanitation infrastructure, and inadequate hygiene practices, which are also addressed in this study.

However, the study makes a valuable local contribution by addressing a critical gap in Aplahoué, Couffo, Benin, where specific conditions of water access (37.4%) and improved sanitation coverage (14.6%) had not been rigorously evaluated before. The evidence provided, particularly on diseases like malaria and diarrhea, identifies localized risk factors such as reliance on wells and rivers, lack of toilets, and a history of previous illnesses in households.

This local relevance is crucial for decision-making and the implementation of targeted policies tailored to the community’s needs.

Thank you for your comment.

  1. What Does It Add to the Subject Area Compared with Other Published Material?

The study fills a local information gap by providing precise data and systematic analysis on the conditions of water access, sanitation, and hygiene in Aplahoué.

It characterizes the specific situation in Aplahoué, highlighting deficiencies in water and sanitation infrastructure, such as low access to potable water (37.4%) and improved sanitation (14.6%). It identifies local risk factors associated with waterborne diseases, such as reliance on wells and rivers, lack of toilets, and the presence of stagnant water.

The study presents an empirical analysis using robust statistical methods (multivariate logistic regression), confirming key associations that allow the design of targeted interventions for this community.

While not globally innovative, the study’s strength lies in its localized approach, which offers practical insights for policymakers and public health programs.

Thank you for your comment.

  1. Are the Conclusions Consistent with the Evidence and Arguments Presented, and Do They Address the Main Question?

The conclusions are consistent with the evidence and arguments presented, adequately addressing the study’s main question.

The conclusions report a prevalence of 45.6% at the household level and 16.6% at the individual level, with malaria and diarrhea as the most frequent diseases.

Significant risk factors are identified, such as a history of previous waterborne diseases in households and hospitalization due to such illnesses.

The findings, supported by robust statistical analyses (Chi-square tests and multivariate logistic regression), confirm that these risk factors significantly increase the likelihood of new disease episodes (OR = 16.99 and OR = 16.39, respectively). The associations between environmental conditions (e.g., stagnant water, lack of toilets) and disease prevalence further strengthen the conclusions.

Thank you for your comment.

Suggestions to improve the conclusions :

Include practical implications for local public policies or intervention programs.

Explicitly state how the results align with or contrast findings from similar studies in other regions.

Acknowledge methodological limitations (small sample size, self-reported data) to enhance transparency and coherence.

We thank you for these relevant suggestions. In line with your recommendations, we have enriched the discussion by including the practical implications of our results for local public policies. We have also further compared our results with those of similar studies conducted in other regions. Finally, the limitations of the study, including the small sample size and the use of self-reported data, have been acknowledged and discussed in order to improve the transparency of the manuscript.

  1. Are the References Appropriate?

The references are appropriate and relevant to the context of the problem:

They include studies conducted in Sub-Saharan Africa (e.g., Benin, Cameroon, Burkina Faso) and authoritative sources like the WHO and UNICEF.

Most references are recent (2016-2022), which is essential for addressing a current issue like waterborne diseases.

Thank you for your comment.

Points à améliorer :

Areas for improvement:

Lack of regional specificity: Although the study addresses gaps in Aplahoué, it does not reference recent studies in neighboring communes of Benin. Including regional studies would strengthen the contextual analysis.

Global comparisons: Additional references from developing regions (e.g., Latin America, rural Asia) would provide a broader global perspective on the issue.

Methodological references: Few studies are cited to support the statistical methods used, such as logistic regression and sampling techniques. Incorporating these would add technical rigor.

Suggested references for inclusion:

Ohwo, O.; Omidiji, A.O. Pattern of Waterborne Diseases in Yenagoa, Nigeria. Journal of Applied Sciences and Environmental Management 2021.

Mebrahtom, S.; Worku, A.; Gage, D.J. The Risk of Water, Sanitation, and Hygiene on Diarrhea-Related Infant Mortality in Eastern Ethiopia. BMC Public Health 2022.

Chen, K.; et al. The Association between Drinking Water Source and Colorectal Cancer Incidence in Jiashan County of China. Eur J Public Health 2005.

Additional suggestion: Review incomplete or general references (e.g., "Smith et al., 2017") to ensure accuracy and clarity.

A literature review was carried out in order to complete the most relevant references.

  1. Comments on Tables and Figures

The tables are clear, well-organized, and provide robust summaries of the key results. However, the visual presentation could be enhanced:

Figures and visual aids: Include bar charts or histograms to illustrate the prevalence of diseases and the distribution of water sources.

Map of the study area: Add a map showing Aplahoué’s location within Couffo, Benin, highlighting data relevant to water access and vulnerability zones. This would provide valuable geographic context.

Interpretation of tables: Summarize key findings in the text, particularly significant variables in the univariate analysis (Table 1) and multivariate results (Table 3).

Highlight significant results: In Table 1, visually emphasize significant variables (p < 0.05) and consider dividing data into thematic sections (demographic, behavioral, environmental, health-related).

Clarify regression results: In Table 3, explain the wide confidence intervals observed for certain variables, as they may indicate low statistical precision.

We thank you for your feedback. In accordance with your suggestions, we have included in the revised manuscript a graph showing the frequency of the diseases studied. A geographical map of the study setting has also been added. The significant results in Table 1 have been visually highlighted. The key takeaways from the results of the univariate and multivariate analysis have been further highlighted in the body of the text in the results section.

The main conclusions regarding the significant variables in Table 1 are presented in section 3.3. Univariate Analysis of the manuscript.

Furthermore, we can say that the wide confidence intervals observed for some variables can be explained by the relatively limited sample size.

Reviewer 2 Report

Comments and Suggestions for Authors

The manuscript reported the prevalence and factors associated with several diseases in southwestern Benin using a cross-sectional study. The study looks interesting and may have important public health significance in that area. However, there are some major issues in the manuscript.

1.     Why are those diseases (malaria, diarrhea, dysentery, typhoid fever, and skin infections) called water-borne diseases? Are there any evidence to show that these diseases are caused due to exposure to contaminated water in that area? As we know, these diseases are not necessarily water-borne. For example, malaria is a vector-borne disease, and diarrhea can be caused by consuming contaminated food.

2.     The data used in this study are mainly based on survey or questionnaire. It is uncertain how those diseases were diagnosed? Were they correctly diagnosed?

3.     The data were collected between September 4 and October 16, 2023, which is less than two months. How was prevalence defined? What is population?

4.     These types of diseases have very different causes and characteristics. As a results, the risk factors can be very different. Why are these diseases grouped together as the dependent variable? It is better to run statistical analysis separately.

5.     The sample size is relatively small, and the survey period is very short.

Minors:

It is better to have a map to show the study area.

In many tables, the number is spelt incorrectly. specifically, comma was misused. For example, ‘3,55’ should be ‘3.55’; ‘1,6 -7,5’ should be ‘1.6 – 7.5’.

Comments on the Quality of English Language

The language of the manuscript needs to be improved.

Author Response

  1. Why are those diseases (malaria, diarrhea, dysentery, typhoid fever, and skin infections) called water-borne diseases? Are there any evidence to show that these diseases are caused due to exposure to contaminated water in that area? As we know, these diseases are not necessarily water-borne. For example, malaria is a vector-borne disease, and diarrhea can be caused by consuming contaminated food.

These diseases are referred to as waterborne diseases because of their potential link to water, either as a direct or indirect vector. They may be associated with exposure to contaminated water or poor hygiene practices due to limited access to quality water. For example, diarrhoea, dysentery and typhoid fever are often caused by the ingestion of water contaminated by faecal pathogens. Skin infections can also occur due to frequent contact with polluted water. In the case of malaria, although it is a vector-borne disease, water plays an indirect but crucial role. The breeding habitats of vector mosquitoes, such as stagnant water, are key factors in transmission.

  1. The data used in this study are mainly based on survey or questionnaire. It is uncertain how those diseases were diagnosed? Were they correctly diagnosed?

Cases were reported by household heads or their representatives. They were asked whether a health worker had diagnosed any of the diseases studied in a household member in the past three months.

  1. The data were collected between September 4 and October 16, 2023, which is less than two months. How was prevalence defined? What is population?

The target population included households in the commune of Aplahoué. Household heads who had given their informed consent and had been residing in the locality for at least six months were included in the study. They were asked whether a health worker had diagnosed any of the diseases studied in a member of the household in the last three months. The waterborne diseases considered were malaria, diarrhea, dysentery, typhoid fever, and skin infections such as scabies. Prevalence was thus defined as the ratio between the number of participants who reported the occurrence of at least one of the diseases studied in the previous three months and the total number of participants included in the study.

  1. These types of diseases have very different causes and characteristics. As a results, the risk factors can be very different. Why are these diseases grouped together as the dependent variable? It is better to run statistical analysis separately.

Previous work has highlighted the existence of common factors associated with the occurrence of these conditions. In this context, the study is part of an integrated approach aimed at identifying household-related determinants that influence the occurrence of these diseases in an environment marked by high prevalence. Unlike a fragmented approach, this integrated strategy promotes more comprehensive, effective and sustainable interventions to reduce the prevalence of these conditions in the communities concerned.

  1. The sample size is relatively small, and the survey period is very short.

The sample size and the period selected for the study were defined according to several organizational and technical constraints. Despite these constraints, significant efforts were made to ensure that the selected participants were representative of the target population. This approach aimed to ensure the reliability of the results and their relevance to the entire study population, taking into account key characteristics and specificities of the context.

Minors:

It is better to have a map to show the study area.

In the revised manuscript, a map was added.

In many tables, the number is spelt incorrectly. specifically, comma was misused. For example, ‘3,55’ should be ‘3.55’; ‘1,6 -7,5’ should be ‘1.6 – 7.5’.

In many tables, the number is spelled correctly. In particular, the comma was used incorrectly. For example, "3 55" should be "3.55-5,"; "1.6 - 7.5" should be "1.6 - 7.5."

Thanks for this observation. We have corrected these errors in the tables, especially the inappropriate use of the comma.

Round 2

Reviewer 2 Report

Comments and Suggestions for Authors

The authors responded some of my concerns, but did not sufficiently answer my first two questions.

1. I did not ask the authors to explain what water-borne disease is. Instead, I asked why exposure to contaminated water is the major cause of these diseases. Can the authors show any evidence? If cannot, the authors need to discuss this as a limitation.

2. For the second question, the authors just repeated the procedure  of data collection, did not answer how to make sure that the data are correct, namely, the diseases were correctly diagnosed. Additional data quality control needs to be mentioned. 

Author Response

  1. did not ask the authors to explain what water-borne disease is. Instead, I asked why exposure to contaminated water is the major cause of these diseases. Can the authors show any evidence? If cannot, the authors need to discuss this as a limitation.

We have no proof and this was presented as a limitation in the discussion.

  1. For the second question, the authors just repeated the procedure  of data collection, did not answer how to make sure that the data are correct, namely, the diseases were correctly diagnosed. Additional data quality control needs to be mentioned

 The second level of control was specified in the method at the level of the technical part and     data collection tools.